# OpenReview forum: "Jailbreaking Multimodal Large Language Models Through Video Prompts"
_ICLR.cc/2026/Conference — ICLR 2026 Conference Withdrawn Submission_

### Official Review · Reviewer_bAxv · 2025-10-23

**Soundness:** 2
**Presentation:** 3
**Contribution:** 2
**Rating:** 2
**Confidence:** 4

**Summary:**

The paper introduces SPTV, a jailbreak method for video large language models. It converts harmful queries and their safe paraphrases into typographic images, uses CLIP-based similarity to select “safe-looking” frames, and assembles them into videos paired with guiding prompts. Experiments show higher attack success rates across models, revealing vulnerabilities in video encoders’ safety alignment.

**Strengths:**

It systematically examines the transfer of image-based jailbreaks to the video domain, empirically revealing weaknesses in video understanding and safety alignment, and introduces a frame-selection framework combined with a strongly guided prompt.

**Weaknesses:**

1. Technically, SPTV appears to be a multi-frame adaptation of existing typographic-based jailbreaks (e.g., FigStep) with an added similarity-driven selection, rather than a fundamentally new attack principle or optimization objective. The authors should compare mechanisms with FigStep and related structured jailbreaks, not only ASR.
2. Frame selection is based on cosine similarity in CLIP’s visual space, but an MLLM’s safety filter may rely on broader cross-modal or overall semantic signals. Correlation between CLIP similarity and refusal-prefix probability is empirical, not causal — the authors need stronger evidence for the claimed causal link.
3. Adopt finer-grained evaluation metrics(e.g., Jailbreaking Multimodal Large Language Models via Shuffle Inconsistency) to more thoroughly measure jailbreak effectiveness and it is better to use GPT4 an automated judge.

**Questions:**

1. Using 1 fps and only 4 frames is far from typical video formats; it’s unclear whether such short, low-fps clips genuinely trigger the model’s video processing pipeline.
2. The proposed method should analyze sensitivity to hyperparameters (frame rate, font size, similarity threshold, etc.).
3. VideoJailPro demonstrates relatively limited performance in the current evaluation. To ensure a fair and compelling comparison, the authors should conduct their evaluation on the same dataset used in VideoJailPro, enabling direct benchmarking under identical conditions.

---

### Official Review · Reviewer_cWe3 · 2025-10-30

**Soundness:** 3
**Presentation:** 2
**Contribution:** 2
**Rating:** 2
**Confidence:** 4

**Summary:**

This paper investigates vulnerabilities of multimodal large language models (MLLMs) in the video modality. The authors show that multimodal jailbreaking attacks can be even more effective when translated into video prompts. Surprisingly, simply stacking identical harmful images into a video sequence increases the attack success rate (ASR) compared to the single-image case. The authors attribute this to differences in representation space: unsafe videos tend to be embedded more similarly to safe ones, resulting in lower refusal probabilities.

Building on this, they propose Safety-Proximal Typographic Video (SPTV) attacks. The method generates diverse-frame typographic videos that interleave paraphrased harmful frames with safety-proximal ones, selected via bipartite matching using CLIP-based cosine similarity. This design makes the video resemble safe data while still encoding harmful content. The method is tested on both open-source and closed-source MLLMs.

**Strengths:**

1. Method simplicity and effectiveness: SPTV is conceptually straightforward but cleverly exploits the gap between video and image safety alignment. The bipartite matching step and typographic paraphrasing build on prior work like FigStep but extend it effectively to video.
2. Comprehensive evaluation: The study benchmarks multiple models and policies, using both open- and closed-source systems, and reports multiple metrics (ASR, feature similarity, log refusal probability).
3. Empirical robustness: Even under system prompt defenses, SPTV maintains relatively high ASR, suggesting generality beyond trivial cases.

**Weaknesses:**

1. Limited novelty in conceptual leap: The idea that multiple harmful frames can have stronger effects than a single frame is somewhat expected. The main contribution is empirical and methodological refinement rather than a conceptual breakthrough.
2. Dependence on typography-based prior work: The method heavily extends FigStep and other image-based typographic jailbreaks; much of the novelty lies in adapting them to video rather than proposing a fundamentally new attack mechanism.
3. Limited interpretability of small probability differences: The refusal log probabilities (e.g., around −6 in Fig. 2b) are already quite small; it’s unclear whether such small differences are practically significant or just statistical noise.

**Questions:**

1. Presentation and formatting issues: Some tables (e.g., Table 2) are dense and difficult to read; improved layout or normalization could help interpret per-policy results.
2. Architectural differences and attack pathways: Different MLLMs process videos differently. For instance, Qwen2.5-VL treats video as a sequence of frame embeddings, while others (e.g., VideoLLaMA2) have explicit spatiotemporal modules. It would be helpful to explicitly analyze whether models with distinct temporal modules exhibit qualitatively different failure modes.

---

### Official Review · Reviewer_p4JK · 2025-10-31

**Soundness:** 2
**Presentation:** 2
**Contribution:** 2
**Rating:** 4
**Confidence:** 3

**Summary:**

This paper explores vulnerabilities in multimodal large language models (MLLMs) by demonstrating that video-based jailbreaking attacks are more effective than image-based ones. The authors propose a novel method, Safety-Proximal Typographic Videos (SPTV), which uses diverse frames to bypass safety mechanisms in MLLMs. Their approach involves augmenting harmful queries, paraphrasing them, and generating typographic videos with frames selected through bipartite matching. The method outperforms previous techniques in jailbreaking popular MLLMs like VideoLLaMA-2 and GPT-4.1, highlighting the need for improved safety in video-processing models.

**Strengths:**

1. The paper is clearly written, with a well-structured presentation that logically progresses from the problem formulation to the proposed solution and experimental results.
2. The paper provides a thorough analysis of the vulnerabilities in MLLMs, including a detailed comparison of image-based and video-based attacks.
3. The paper introduces a novel perspective on jailbreaking MLLMs by shifting focus from image-based attacks to video-based attacks. While image-based jailbreaking has been well-studied, the authors' exploration of video-based attacks and their discovery that stacking identical frames into a video format can bypass MLLM safety mechanisms is an original and insightful contribution.

**Weaknesses:**

1. While the paper demonstrates how the SPTV method can bypass safety mechanisms in MLLMs, it does not explore or evaluate potential defense mechanisms specifically designed for video-based attacks. The paper could be strengthened by investigating how MLLMs could be made more resistant to such attacks, either through better safety alignment in video processing or through the use of adversarial training techniques.
2. Although the paper evaluates the attack success rate (ASR) across 16 different safety policies, it does not provide a detailed analysis of which policies are more susceptible to video-based attacks or why. A deeper dive into the specific characteristics of the safety policies that fail most often under the SPTV attack would help in understanding the weaknesses in safety design.
3. Ambiguity in the analysis of frame diversity and repetition. The paper first states that repeated frames are more similar to those in safe videos, making them harder to detect by the model’s safety mechanisms. However, it later claims that diverse frames are considered more effective for attacks. These two conclusions contradict each other.

**Questions:**

Please refer to the weaknesses.

---

### Official Review · Reviewer_WQxP · 2025-11-04

**Soundness:** 3
**Presentation:** 3
**Contribution:** 2
**Rating:** 4
**Confidence:** 3

**Summary:**

This paper reveals that multimodal large language models are more vulnerable to video-based jailbreaking than image-based attacks, as unsafe videos are embedded closer to safe ones in the representation space.
It proposes Safety-Proximal Typographic Videos (SPTV), which interleave paraphrased harmful text with visually diverse, safe-looking frames selected via bipartite matching, effectively bypassing safety filters.
Across multiple models (e.g., VideoLLaMA-2, Qwen2.5-VL, GPT-4.1, Gemini-2.5), SPTV achieves state-of-the-art attack success rates, highlighting the urgent need for stronger video-specific safety alignment.

**Strengths:**

The paper introduces a novel attack vector—video-based jailbreaking—revealing vulnerabilities in multimodal large language models and proposing the innovative Safety-Proximal Typographic Video framework that bridges temporal and visual modalities.

The methodology is rigorous, combining embedding-space analysis, quantitative evaluations, and algorithmic optimization (via Hungarian matching) to substantiate claims with solid empirical evidence across both open- and closed-source MLLMs.

The presentation is clear and well-motivated, with intuitive visualizations and ablation studies, and the findings have substantial implications for AI safety and alignment, especially as video-capable foundation models become widespread.

**Weaknesses:**

The work convincingly exposes vulnerabilities in video-based MLLMs, but it would be helpful if the authors could discuss possible defense implications. For example, do the observed embedding-space similarities suggest new directions for video-level safety alignment or temporal consistency checks to mitigate such attacks?

If using multiple images/video prompts to attack the model, wouldn't that be easier to be detected compared to image input? Besides ASR, can you test some jailbreak detection framework on your model?

The paper uses Llama-Guard to automatically label outputs as safe or unsafe when computing ASR, but it does not report the detector’s true/false positive rates. This introduces potential bias in ASR measurement.

It is not stated whether the attacker has white-box or black-box access, how many queries are required, or whether video encoders are shared between models

The paper could better justify design choices (e.g., number of frames K, fps, matching algorithm) through controlled ablations

**Questions:**

Same as weaknesses.

---

### Note · Authors · 2025-11-14

I have read and agree with the venue's withdrawal policy on behalf of myself and my co-authors.